

# Choose your diffusion: Efficient and flexible ways to accelerate the diffusion model in fast high energy physics simulation

**Cheng Jiang**[1⋆]**, Sitian Qian**[2†] **and Huilin Qu**[3‡]

**1** School of Physics and Astronomy, University of Edinburgh,
EH9 3FD, Edinburgh, United Kingdom
**2** School of Physics, Peking University, 100871, Beijing, China
**3** CERN, EP Department, CH-1121 Geneva 23, Switzerland

⋆ C.Jiang-19@sms.ed.ac.uk , † stqian@pku.edu.cn , ‡ huilin.qu@cern.ch ,

## Abstract

The diffusion model has demonstrated promising results in image generation, recently becoming mainstream and representing a notable advancement for many generative modeling tasks. Prior applications of the diffusion model for both fast event and detector simulation in high energy physics have shown exceptional performance, providing a viable solution to generate sufficient statistics within a constrained computational budget in preparation for the High Luminosity LHC. However, many of these applications suffer from slow generation with large sampling steps and face challenges in finding the optimal balance between sample quality and speed. The study focuses on the latest benchmark developments in efficient ODE/SDE-based samplers, schedulers, and fast convergence training techniques. We test on the public CaloChallenge and JetNet datasets with the designs implemented on the existing architecture, the performance of the generated classes surpass previous models, achieving significant speedup via various evaluation metrics.



# 1   Introduction

The Large Hadron Collider (LHC) [1] is one of the biggest particle accelerator human have ever made. More than billions of events are generated and recorded by experiments like ATLAS [2] and CMS [3] near the interaction points during each run. Sufficient and precise simulation for comparing the data and theoretical prediction is required to find the evidence of new physics at the energy frontier.

The `Geant4` software toolkit [4] is applied for the detector simulation. The simulated primary particles would propagate through specific material and geometry, resulting vast amount of secondary particles produced in each propagated steps to achieve precision. As a consequence, the detector simulations for the modern high granularity detectors occupy the most computation resources [5]. Reconstructing the physics objects like jets with complicated substructure for different physics process also plays a crucial role in providing the enough statistics for the data-driven analysis. In the upcoming High Luminosity LHC [6], the expected integrated luminosity will scale up to ten times the current level. It is nearly impossible to perform the full simulation for all the events with constrained computational budget, manifesting the necessity of fast simulation.

To address with these challenges comprehensively, different techniques adopted from deep generative modeling, where a lot of phenomenal improvements have quickly emerged over the years, have become the popular approach as fast surrogate models in high energy physics.

One latest category of generative models, the Diffusion Model (DM) as well as its variations, has become increasingly dominant in various generative applications [7–21]. There has been a few studies using DM in both fast detector and end-to-end event simulations.

For fast detector simulation, `CaloScore` [22] utilizes the score-based generative model and the conditional UNet to produce almost indistinguishable generated showers from truth ones. `CaloDiffusion` [23], on the other hand, utilizes the denoising diffusion probabilistic model, incorporates novel geometry adaptation, and enhances the UNet with middle self-attention and cylindrical convolutional layers. This combination positions `CaloDiffusion` as one of the top models in fast detector simulation. Moreover, `CaloClouds` [24] offers an alternative way to generate geometry-independent point clouds for great modelling of physically relevant distributions.

For end-to-end event simulation [20, 21], there are a series of studies for generating the point cloud with DM [25, 26]. Notably, those studies also use architectures like transformer [25] or `EPiC` layer [26] that utilize the permutation invariance of jet constituents.

Among those original studies, their first versions often suffer from a long generation time, as desired results typically converge only after a large number of sampling steps. Further studies involving the fresh distillation methods like progressive distillation [27, 28], consistency

model [23, 29, 30] to get the result in only few steps with the exchange of longer training time, and other ways [29, 31] to speedup the backward process.

Most of the time, different DM methods face a tradeoff between the quality and speed of generated classes. There is a lack of a comprehensive study on the training and sampling dynamics of diffusion models applied for various purposes and datasets in high energy physics. In this study, we investigate the effects of different training strategy by offering a soft way to expedite the convergence of performance. We also adopt several popular training-free ODE/SDE based samplers and noise schedulers in post-DDPM era [32–36]. By balancing the discretization errors and accumulated errors in the sampling process, a flexible and efficient way can be offered to make the diffusion model faster and more accurate.

We finally make our approaches portable by testing in different network and different datasets using the public fast and high-fidelity calorimeter dataset *CaloChallenge* [37, 38] and the jet point cloud dataset *JetNet* [39]. The performance surpasses the previous model by several evaluation metrics [8, 25, 37, 38, 40].

This paper is outlined as follows, Section 2 covers the fundamentals of diffusion model, elucidating how data such as shower images or point clouds are specifically generated through the backward process. Section 3 provides the a brief introduction about the dataset used in this study. Section 4 discusses the training strategy applied. Various metrics will be presented to evaluate the performance in Section 5. Lastly, Section 6 gives a empirical discussion, conclusion and outlooks about the study.

## 2 Dynamics in the diffusion model

The diffusion model was initially introduced to sample complex data distributions by drawing inspiration from concepts in non-equilibrium thermodynamics [41]. At first, we gradually add noises to an initial data sample from $x_0$ to $x_T$ until the time step $T$, usually denoted as the forward process $q$ where the data points are degraded as $T$ increase. Then it follows a backward process $p$ where noisy data could be denoised iteratively to new observation $\sim x_0$ as the time goes reversely.

Two main classes are introduced later [42, 43]: Denoising Diffusion Probabilistic Models (DDPMs) [44, 45] and Score-based Generative Models (SGMs) [46].

For DDPM, the forward process transforms input distribution into a prior distribution by

$$q(x_t|x_{t-1}) = \mathcal{N}(x_t|\sqrt{1-\alpha_t}x_{t-1}, \alpha_t), \tag{1}$$

where $\alpha_t$ is the noise variance in every time step in case of the noise has unit variance $\epsilon \sim \mathcal{N}(0, \mathcal{I})$, the single step formulation can be rewritten as:

$$q(x_t|x_0) = \mathcal{N}(x_t|\sqrt{\tilde{\alpha}_t}x_0, 1-\tilde{\alpha}_t), \tag{2}$$

$$x_t = \sqrt{\tilde{\alpha}_t}x_0 + \sqrt{1-\tilde{\alpha}_t}\epsilon, \tag{3}$$

where $\tilde{\alpha}_t = \prod_{i=1}^{T}(1-\alpha_i)$. The direct calculation from $x_0$ can allow us to random sample time steps to add noise during training. In the backward process, a probability function $p(x_{t-1}|x_t)$ is learned to denoise the noisy data at previous time step until reach the initial input $x_0$.

The SGM is formularize similarly, in the score Stochastic Differential Equations (SDEs), the forward process is defined by [46]:

$$dx = f(x,t)dt + g(t)w_t, \tag{4}$$

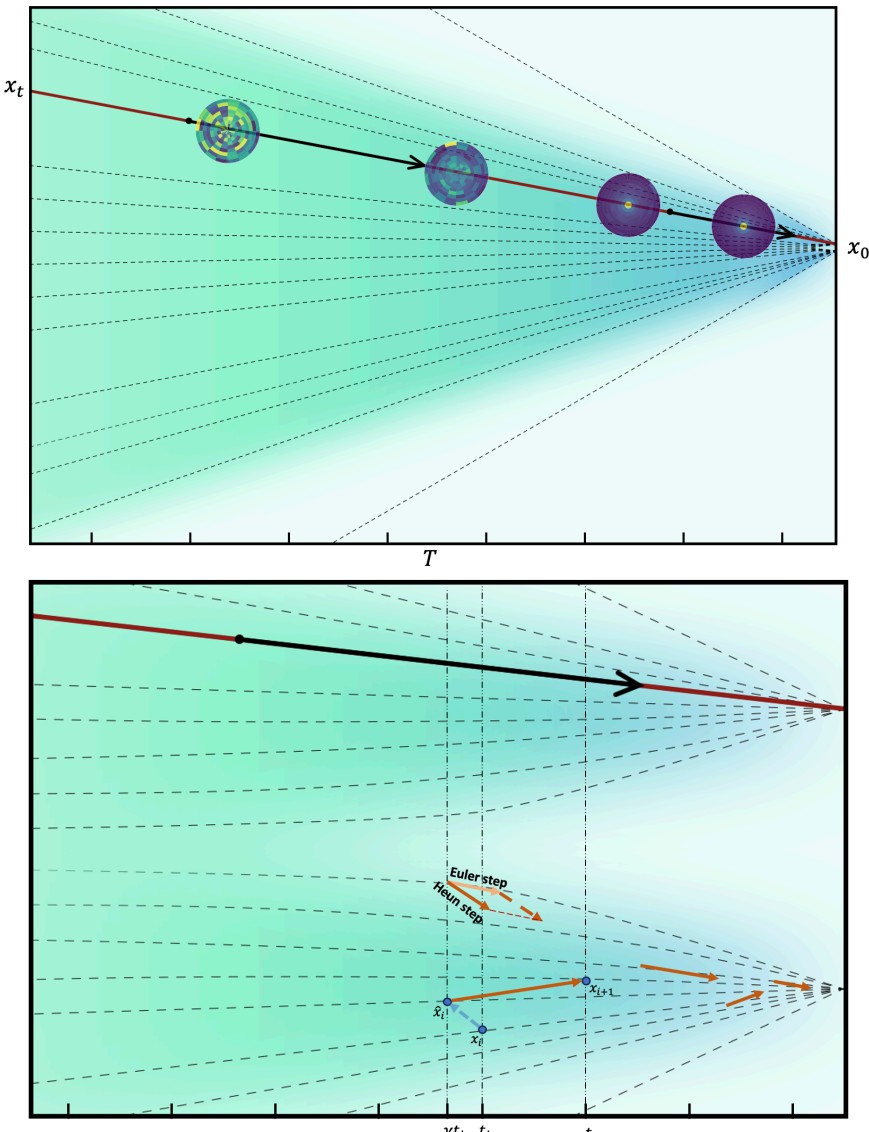

Figure 1: Upper: The schematic diagram for a noisy shower image (energy deposited in one layer of calorimeter) $x_t$ denoises along the ODE curvature with varying time steps to clean shower close to $x_0$. Lower: Arrows indicate a schematic sketch how ODE and SDE samplers reach the $x_0$ along timesteps, ODE flows along the certain direction toward $x_0$. $t_i$ is the $i^{th}$ timestep, SDE will have extra noise injection $\gamma t_i$ in each time step before moving to next step. The Euler and Heun are the first and second order calculation for updating $x_i$.

where $f$ and $g$ are drift and diffusion coefficients chosen differently for variance preserving and variance exploding noise, $w$ is the standard Brownian motion.

The reverse-time SDE is defined as:

$$dx = [f(x, t) - g(t)^2 \nabla_x \log p_t(x)]dt + g(t)d\tilde{w}, \tag{5}$$

unlike DDPM, the quantity we are solving is the score function (A vector field indicating the direction toward regions of increased data density given a certain level of noise.) $\nabla_x \log p_t(x)$. The learning process of this gradient of log probability functions offers another possibility to use not only SDE but also ODE to get the solution.

Then follows by the generalization from [47], we can write the reverse SDE equation as a sum of probability flow ODE and Langevin diffusion SDE:

$$dx = -\dot{\sigma}(t)\sigma(t)\nabla_x \log p_t(x)dt - \beta(t)\sigma(t)^2\nabla_x \log p_t(x)dt + \sqrt{2\beta(t)}\sigma(t)dw_t, \tag{6}$$

where $\sigma(t)$ is the noise schedule, $\beta(t) = \dot{\sigma}(t)/\sigma(t)$, the first term is the marginally equivalent ODE, the second and third terms are added as the deterministic flow and noise injection term for Langevin diffusion, the schematic diagram as shown in Fig. 1.

This determines the characteristics of the general ODE/SDE solver. ODE solvers often have smaller discretization errors than SDE solvers in smaller sampling steps [48, 49]. However, as the sampling steps become larger, the accumulated error for SDEs would be much smaller than that for ODEs, resulting in a smaller total error (better performance) in large sampling steps. The noise scheduler determines how fast those samplers should learn along different time steps. In the paper, we will cover up to seven mainstream samplers nowadays (four ODEs and three SDEs) over different schedulers.

The schedulers and samplers directly decide how the noisy data $x_t$ will reach $x_0$ with proper noise levels. Our choices for schedulers and samplers are listed belows:

- Schedulers:

    - The default cosine beta scheduler for discrete time DDPM pretrained model, this is the baseline to compare with performance for other schedulers and samplers. $\bar{\alpha} = \cos(\frac{\pi(t/T+s)}{2(1+s)})$ where $s = 0.008$ that determines the minimum noise for last steps. We can tell by the function, the noises are added by a linearly decrease rate during intermediate steps to avoid too much noise in the last few steps.

    - Variance Preserving schedulers as a more generalized version of cosine beta schedulers used in both score matching [46] and DDPM, defined as a continous time step $t$ and $\sigma = \sqrt{(\exp\frac{\beta_{max}t^2}{2} + \beta_{min}t - 1)}$ where $\beta_{min}$ and $\beta_{max}$ are constant to determine the lower and upper bound of noise level.

    - The schedulers proposed by Karras [47]. $\sigma_{i<N} = (\sigma_{max}^{1/\rho} + \frac{i}{N-1}(\sigma_{min}^{1/\rho} - \sigma_{max}^{1/\rho}))^\rho$ and $\sigma_N = 0$. It's the monotonically decreasing function, so the noise added in a decreasing rate, $\rho$ determines the curvature. Larger $\rho$ means more steps near $\sigma_{min}$ are shortened. Reasonable value for $\sigma_{min}, \sigma_{max}, \rho$ in our study should around $[0.01, 0.001], [20, 80], [5, 10]$ respectively. This scheduler is helpful for generating good quality of data in low sampling step region.

    - Combine Karras schedulers proposed with Lu [50] by transforming the $\sigma$ to log scale. $\sigma_{i<N} = (\log(\sigma_{max})^{1/\rho} + \frac{i}{N-1}(\log(\sigma_{min})^{1/\rho} - \log(\sigma_{max})^{1/\rho}))^\rho$. The values for $\sigma_{min}, \sigma_{max}$ remains the same region, $\rho = 1$ to keep a healthy noise level range. This scheduler is even more decreasing than Karras schedulers, so the step size changes more rapidly in low sampling step while more slowly in last steps.

- Samplers:

    - The same DDPM samplers are used in [23,44]. The Euler-Maruyama method approximates the reverse diffusion process by iteratively denoising the sample while adding stochastic noise at each discrete timestep.

    - EDM Stochastic sampler with Heun's method [47], as illustrated in Fig. 1, a noise injection step is performed to current time step $\hat{t}_i = t_i + \gamma t_i$ where $\gamma$ is determined by three parameters $\gamma = \min(S_{churn}/N, \sqrt{2} - 1)$ when $S_{\min} \leq t_i \leq S_{\max}$. The $S_{\max}, S_{\min}$ is the range for injection, $S_{churn}$ is the stochasticity strength which give the lower bound to start adding noise. It controls how fast the sampler converges to zero noise. The optimal sets of parameters can make the sampling faster and better. Contrarily, a bad sets of those stochastic term will make possible degradation of the generated sample. In our study, such degradation only happens when extreme values (e.g. large $\gamma$, or $S_{churn}$ close to initial timestep) are selected. The difference between this approach and Euler Maruyama is that the denoising steps in EDM samplers happens in some intermediate state $\hat{t}_i$ instead of actual timestep state $t$, the difference between two states become diminishing when N goes large. This SDE sampler, unlike other SDEs that only give satisfied result after a very large sampling steps, can converge quickly if one can manage to tune those stochastic parameters well.

    - Restart sampler proposed in [48]. This sampler introduces a way to strongly contract the accumulated error based on EDM ODE Heun sampler. A back-and-forth ODE step will repeat $K$ times in the predefined single or multiple intervals $[t_{\max}, t_{\min}]$.

    - DPM-Solver samplers proposed in [49,50]. A popular ODE solver that can speedup the sampling process by applying change-of-variable to noise level. We choose to use the latest multistep solver that calculate iteratively by the knowledge of current and next timestep.

    - Combined DPM-Solver and EDM SDE samplers. Instead of using Euler and Heun step, it uses the "mid-point" step by change-of-variable to intermediate state $\hat{t}_i$.

    - LMS sampler [51]. For each iteration, the sampler will denoise the input data using the model and updates the $x_i$ using the linear multistep integration method with the calculated coefficients of predefined orders.

    - Uni-PC sampler introduced in [52]. A unified predictor-corrector framework that can be applied on many existing samplers. We apply it with DPM-Multistep-sampler with both varying coefficient and $B(h)$ solver proposed in the paper.

## 3 Dataset

### 3.1 CaloChallenge

Fast Calorimeter Simulation Challenge 2022 [37] is the GEANT4 simulation for particle showers deposited in the calorimeter. The purpose for this challenge is to develop and benchmark the fast and high-fidelity calorimeter simulation using modern techniques from deep learning. Three datasets are provided ranging from easy to hard. The complexity of dataset is determined by the dimensionality of the calorimeter showers, including factors such as the number of layers and voxels on each layer.

In each dataset, particles are simulated to propagate along the z-axis inside the cylindrical and concentric detector. Similar to the sampling layers in the common calorimeter, a discrete

number of layers are placed with specific radial and angular bins $r$ and $\alpha$. Two classes of quantities are stored: the incident particle energies and voxel energies stored in each shower. By mapping the voxel location with given $r$ and $\alpha$ bins, one can reconstruct the layer deposited energy inside the shower. This offers flexibility for designing the generative modelling tasks using either low or high level features.

This study will primarily focus on testing dataset 2 which simulates a physical detector with 45 layers of absorber material (Tungsten) and active material (Silicon). A total of 100,000 electrons in each file enter the calorimeter with a direction. The incoming particle energies span over a log-uniform distribution from 1 GeV and 1 TeV. Each layer contains 16 angular and 9 radial bins, so 16×9 = 144 cells, and a total number of 144×45 = 6480 voxels in each shower. A minimal cutoff around 15 keV is applied to voxel energy.[1]

### 3.2 JetNet

*JetNet* [39] is the point cloud like dataset which consists of 200,000 simulated events. Each event contains the corresponding jet types with transverse momenta $p_T \sim 1$ TeV, originating from either gluons, top or light quarks. Additionally, a narrow kinematic window is imposed, ensuring that the relative transverse momenta of the generated parton and gauge boson $\Delta p_T / p_T \leq 0.1$. The jet clustering is performed using the anti$-k_T$ algorithm [53] within a cone size of $R = 0.4$.

For up to 30 particle constituents stored in one jets by descending $p_T$ order, three features are provided: the relative angular coordinate of particles w.r.t. the jets: $\eta^{\mathrm{rel}} = \eta^{\mathrm{particle}} - \eta^{\mathrm{jet}}$, mod ($\phi^{\mathrm{rel}} = \phi^{\mathrm{particle}} - \phi^{\mathrm{jet}}, 2\pi$) and relative transverse momentum $p_T^{\mathrm{rel}} = p_T^{\mathrm{particle}} / p_T^{\mathrm{jet}}$. There is also a fourth variable that masks whether the particles is genuine or zero-padded when there are fewer than 30 constituents stored in the jets.

We will show preliminary results for testing on gluon and top quark jets. The gluon jet can serve as a baseline test since it usually produces the dominant background for many physics processes and has a relatively simple topology. The top jet has more subtle substructure and presents challenging tasks for generative models.

## 4  Preprocessing and training

In this section, we provide an overview for the data processing, and training details on two datasets mentioned in Section 3.

### 4.1  CaloChallenge

We follow a common preprocessing procedure for *CaloChallenge* dataset.

- First, the voxel energies $v_i$ are normalized by the incident particle energy.

- Then a logit-norm transformation is applied to ensure input data has zero mean and unit variance.

$$u_i = \log\left(\frac{x}{1-x}\right), \quad x = v_i - 2\delta v_i + \delta,$$
$$u_i' = \frac{u_i - \bar{u}}{\sigma_u}. \tag{7}$$

Here, $\delta = 10^{-6}$ is introduced to avoid the discontinuity.

---

[1]The rationale behind selecting this dataset is that previous studies indicate it exhibits a comparable level of difficulty to the more finely granulated dataset 3, all while achieving training and inference speeds that are ten times faster.

We choose to use the network from [23] which is composed of a conditional UNet with middle layer self-attention layer, cynlindircal convolution and Geometry Latent Mapping (GLaM). The latter two techniques are found to be beneficial for generating quantities in $r$ and $\alpha$ bins. In order to demostrate the portability of our studies, the neural network architect remains unchanged from the version in the literature [23]. The original training is tailored for discrete noise level samplers like DDPM [44].We have integrated our training setups, including different loss function designs and data preprocessing, into the new training process. We use publicly available pre-trained models with 400 DDPM steps as the baseline to compare the performance.

## 4.2  JetNet

For *JetNet* dataset, we follow the preprocessing procedure from [25, 26]. The Equivariant Point Cloud (EPiC) layer based on the deep sets framework is chosen to be used as it shows great quality of generated classes while having lighter architecture than transformer-based models [26]. With the same training procedure changes, we further modify the architectures with similar sinusoidal and cosine positional embedding by leveraging sine and cosine functions of the noise levels.

Training the diffusion model is essential for generating higher quality showers across different noise levels. We delve into two questions: 1. How can the data with added noise be correctly weighted and learned by the model so it can be denoised more easily in the backward process? 2. How can the training converges faster to a better reliable result?

## 4.3  Training details

We follow the way from [47] to rewrite the score function to a denoiser function which minimizing the $L^2$ denoising error between $x$ and model output $D(x)$ at each noise level:

$$\nabla_x \log p_\sigma(x) = (D(x) - x)/\sigma^2.$$ (8)

If the network is preconditioned with $\sigma$ independent skip connections, $D$ can be written as $D_\theta(x, \sigma) = c_{\text{skip}}(\sigma)x + c_{out}(\sigma)F_\theta(c_{in}(\sigma)x, \sigma, z)$. A set of preconditioning parameters used to predict noise $\epsilon$ with this denoiser function:

$$
\begin{aligned}
\mathcal{L} &= \mathbb{E}_{\sigma, \epsilon}\left[w(\sigma)\|D_\theta(x, \sigma) - x_0\|_2^2\right] \\
&= \mathbb{E}_{\sigma, \epsilon}\left[w(\sigma)\|F_\theta(c_{in} * x_\sigma, \sigma, z) - \frac{1}{c_{out}}(x_0 - c_{\text{skip}}(\sigma) * (x_\sigma))\|_2^2\right],
\end{aligned}
$$ (9)

where $F_\theta$ is the preconditioned network, $c_{in}, c_{out}$ maps the input and output data magnitude, and $c_{\text{skip}}$ controls the skip connection of the network. The parameters $c_{in} = 1/\sqrt{\sigma^2 + \sigma_c^2}$ and $c_{out} = (\sigma * \sigma_c)/\sqrt{t^2 + \sigma_c^2}$, $\sigma$ is the current noise levels, $\sigma_c$ is the constant value, a reasonable value is around $[0.5, 1.5]$ to avoid large variation for gradient scale. $c_{\text{skip}} = \sigma_c^2/(\sigma^2 + \sigma_c^2)$ is important quantity to partially mask the skip connection when the noise level goes large. This can prevent the network from inaccurate prediction. The setting is important in our case given noise varies widely in most of samplers we show later. When the $\sigma$ goes higher, the $c_{\text{skip}}$ becomes smaller. In a way, this compels the skip connection to be deactivated, encouraging the network to prioritize learning the signal for overriding the input rather than predicting noise to correct it. We choose the continuous timestep training where $\log t = \mathcal{N}(P_{\text{mean}}, P_{\text{std}}^2)$ where $P_{\text{mean}}, P_{\text{std}}$ is -1.2 and 1.2 correspondingly.

From our study, the choice of the weighting function is very important as it not only determines the speed of loss convergence but also the quality of the outputs and how the specific energy distribution tends to evolve across different time steps. To mitigate the instability of

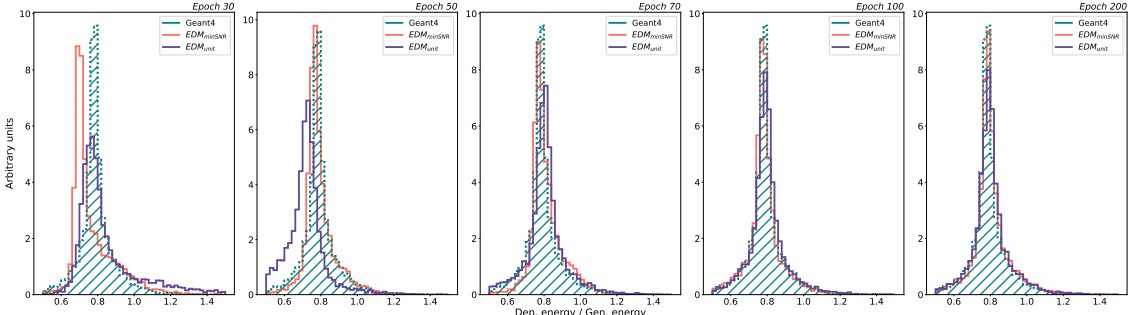

Figure 2: The distribution for total shower energy divided by the incident particle energy, tale shaded region is the GEANT4 reference, the solid line is the generated distribution by 79 steps EDM samplers with different weight at 30, 50, 70, 100, 200 epochs. Purple: default denoiser training with unit weight. Pink: proposed min-SNR weighting. The default training still struggles to learn this variable after 200 epochs whereas the training with proposed weighting strategy already gives good shape alignment after 100 epochs.

the optimization over the training, and balance the gradient magnitude over diverse noise levels, one loss weight category is often mentioned — Signal to noise ratio (SNR) weighting [32,54–56]. We choose the min-SNR weighting as it is found to be beneficial for training high-resolutions dataset. The original min-SNR weighting proposed from [57] defined to be $\min\{\text{SNR}(t),\gamma\}$, where SNR(t) is defined as $\frac{1-\sigma^2}{\sigma^2}$, $\gamma$ is the upper limit for weighting value, usually set to be 2-5. This could avoid the model putting too much weight on the low noise level.

In stead of directly clamping a constant value, we propose a softer version of min-SNR weighting which is more flexible to find the optimal value for better training. Specifically, the weight function is

$$w(t) = (\sigma * \sigma_c)^2/(\sigma^2 + \sigma_c^k)^2, \tag{10}$$

that $\sigma$ and $\sigma_c$ are the same as the preconditioning parameters, the reasonable value of k in the range of $[2, 4]$. This function has one minimum value of 0 around zero $t$ and two local maxima on positive and negative t-axes. The constant k determines the local maximum weight value, and the SNR value where the maxima start to smoothly fall down to 0. The advantages of this weighting scheme will be shown later in Sec 5 with comparison of default constant (unit) weighting strategy.

## 5 Performance

Various schemes for the backward diffusion process have been discussed in previous section. To comprehensively compare their performance and enhance understanding of choosing the appropriate samplers and schedulers for a specific purpose, we offer several methods for evaluation.

### 5.1 CaloChallenge

There are multiple important distributions related to the layer and voxel energies for *CaloChallenge* dataset. We compare the generated samples with GEANT4 as reference. The voxel energy distribution is defined as the distribution of raw generated low level voxel energies. The total energy distribution is defined as the distribution of total energy of the generated shower. The

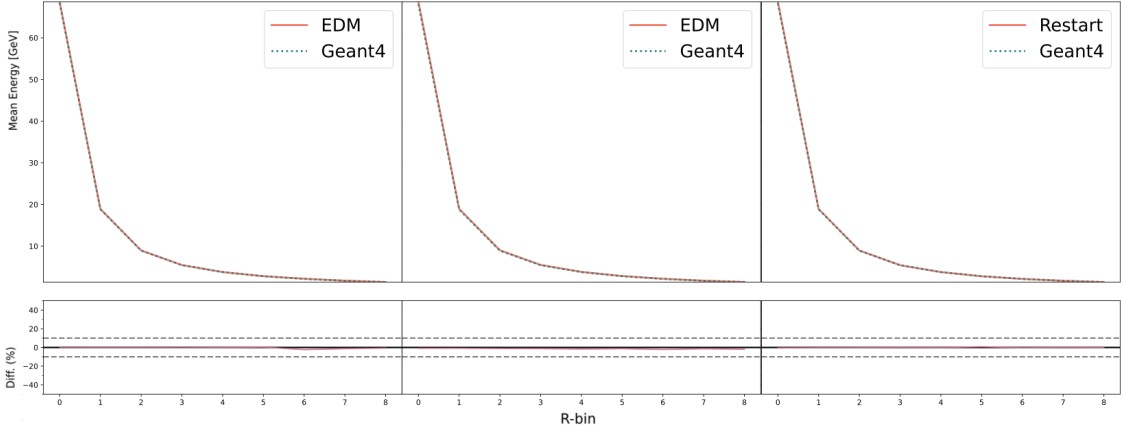

Figure 3: Comparison plots for average center energy of 5000 showers in radial bins. left: EDM with 30 steps, middle: EDM with 79 steps, right: Restart with 30 steps. Those samplers are chosen because of their good sample quality and inference speed trade-off, while the difference almost indistinguishable.

center energy of the shower is defined as $\bar{x} = \frac{\langle x_i E_i \rangle}{\sum E_i}$ for cell location $x_i$ and energy $E_i$, and shower width is defined as $\sqrt{\bar{x^2} - \bar{x}^2}$, we will show these two quantities at the polar coordinate $\alpha$ and $r$. The mean layer energy distribution defined as the mean of generated layer energies. The shower response $E_{\text{ratio}}$ is the total shower energy divided by the incident particle energies, this quantity is extremely important for energy calibration of many particles at LHC. As it will be used to correct the deposited shower energy inside calorimeter into the real particle energy, the small difference between generated sample and GEANT4 means more consistency for particle reconstruction procedure of both fast and full simulation. We will also evaluate the separation power between generated $E_{\text{ratio}}$ and reference one.

In addition, a public evaluation network [37] DNN classifier with 2 layers, each comprising 2048 nodes and a 0.2 dropout rate, is employed to assess the area under the receiver operating characteristic curve (AUC) for all high-level features, including center energies, shower width, and layer energies, of both generated samples and references. An AUC value that close to 0.5 suggests that the generated and target classes are nearly indistinguishable. Conversely, a higher AUC value indicates a more evident distinction between the quantities of the two classes.

First, a comparison plot for $E_{\text{ratio}}$ with default EDM training loss weight and our proposed training weight is shown in Fig. 2. The choice of the weighting function influences how the kinematics distribution evolves over time. After 200 training epochs, the model trained with the default weighting still struggles to learn the $E_{\text{ratio}}$ distribution. In contrast, employing proposed min-SNR weighting results in a better alignments with reference histogram, the generated $E_{\text{ratio}}$ aligning with GEANT4 sample after only 100 epochs. This observation affirms that min-SNR weighting expedites the convergence of high-level features to achieve better performance compared to constant weighting.[2]

Few plots for center energy in $r$ direction are selected in Fig. 3. We choose the EDM Heun SDE and Restart samplers that have good sample quality and inference speed trade-off. Notably, the EDM samplers and Restart samplers with Karras schedulers can learn this

---

[2]A 1D layer diffusion can generate better $E_{\text{ratio}}$ distribution. But it might be over simplified as the model only considers the layer deposited energy. The paper only considers the low level voxel diffusion as a more realistic and challenging case for fast calorimeter simulation.

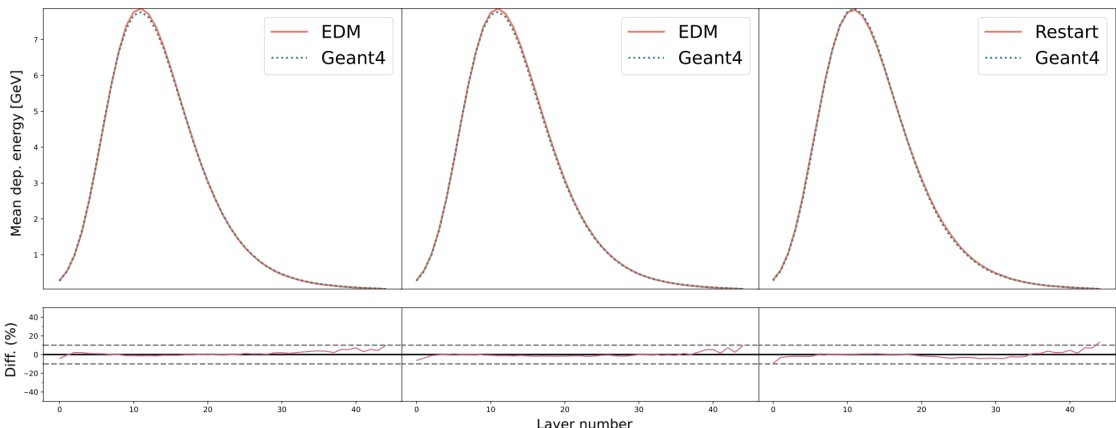

Figure 4: Comparison plots for mean layer energies of 5000 showers. left: EDM with 30 steps, middle: EDM with 79 steps, right: Restart with 30 steps.

quantities really well. The ratio difference is almost indistinguishable with naked eyes. The default scheduler for the EDM sampler is Karras. To better illustrate the impact of different noise schedulers on the same sampler, a broader exploration of scheduler choices—such as Karras, DPM++, and Lu schedulers with varying decay rates for $\sigma$ is conducted. For small ($\sim$30) steps EDM, the differences in last three radial bins are larger meaning it underestimates the lower energy region. When the steps become larger, we see the EDM predicts well in all radial bins. The Restart sampler can achieve this with only 30 steps. Given the Restart samplers works with a predefined parameters for configuring the EDM ODE, the generation time is directly proportional with the restart iterations in the configuration, usually longer than default EDM samplers. A lack of principled way for these hyperparameters is the inherent limitation for this sampler. Later we will see the best performance is achieved around 30-70 steps as we only use specific configurations in this range.

Then the mean layer energies, total shower energies, shower response, and voxel energies for the same three cases are shown in Fig. 4, 5, 6.

The low steps EDM sampler has good performance on low energy voxel while mediocre performance on high energy tails of voxel distributions. In Fig. 5, we have seen the shape of shower response between generated and reference sample for Restart and EDM samplers are more aligned than other samplers, especially around the peak of histogram ($\sim$0.8). DPM-solver sampler with Karras scheduler does very well on predicting the voxel energy around 15 steps, achieving an exceptional agreement on both low and high energy tails shown in the left plot of Fig. 6. The energy thresholds also affect low voxel energies generated by different samplers. For the baseline comparison, we used the pre-trained CaloDiffusion [23] model for dataset 2 with default 400 steps DDPM has 4-10% difference in the intermediate energy level (5-1000 MeV), and less than 30% difference with GEANT4 until 0.7 MeV. A clear difference has been seen below 0.7 MeV. The same scenario happens to EDM, Restart and LMS samplers, but all three samplers show more robust performance (i.e. a smaller excess in low energy tail). Generated showers produced by EDM and Restart samplers both have much less difference with GEANT4 around the intermediate energy level. Two new ODE samplers, DPM-Solver and UniPC, hold only less than 10% difference in low energy region, nevertheless perform worse on other regions. Indeed, some of them are struggling to match the low voxel energy, the presence of the tail is probably a consequence of model itself and too low energy threshold. It is expected this deficiency should not have big impact on the downstream reconstruction of shower.

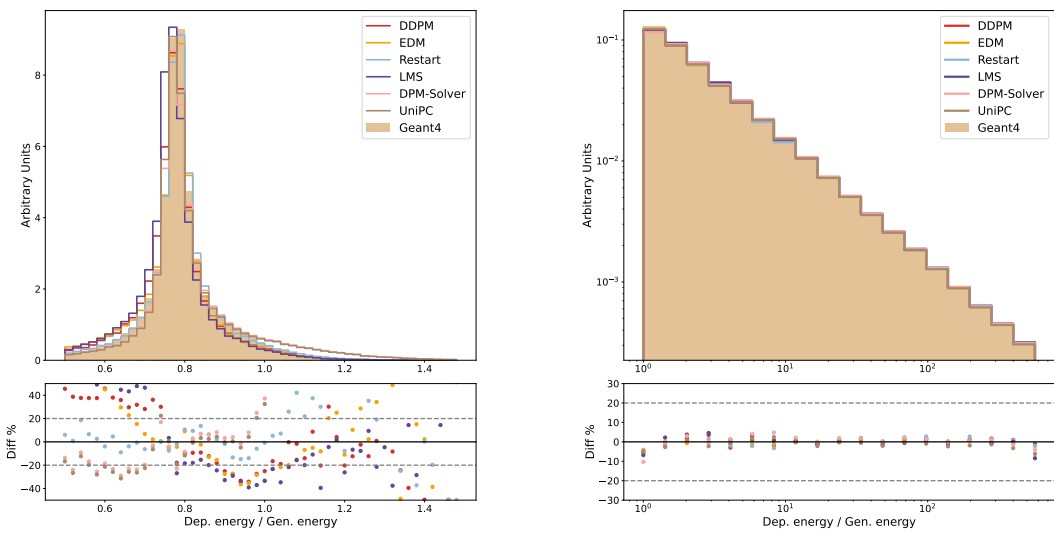

Figure 5: Comparison histograms for 100k shower energies. left: shower response, right: total shower energy from 400 steps DDPM (CaloDiffusion), 79 steps EDM, 30 steps Restart 36 steps LMS, 15 steps DPM-Solver ++, and 15 steps UniPC samplers.

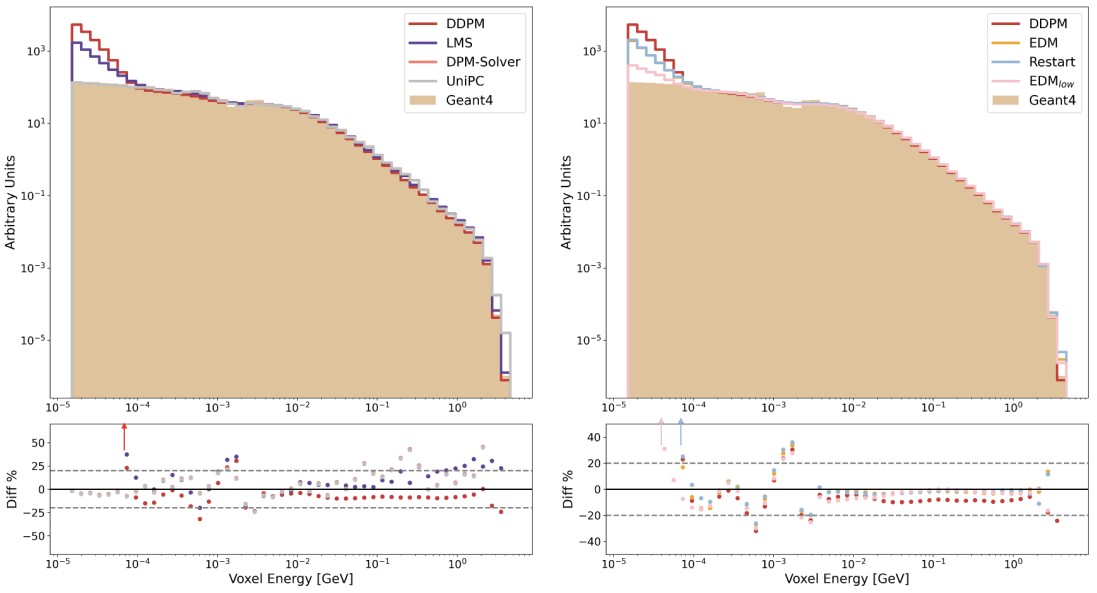

Figure 6: Comparison histograms for raw voxel energies of 100k showers: Left: 30 steps EDM, 79 steps EDM, 30 steps Restart samplers; Right: 400 steps DDPM (CaloDiffusion), 36 steps LMS, and 15 steps DPM-Solver ++ samplers. Bottom is the percentage of difference between Geant4 and generated samples. The upward arrows indicate that the differences fall beyond the window at certain voxel energy levels.

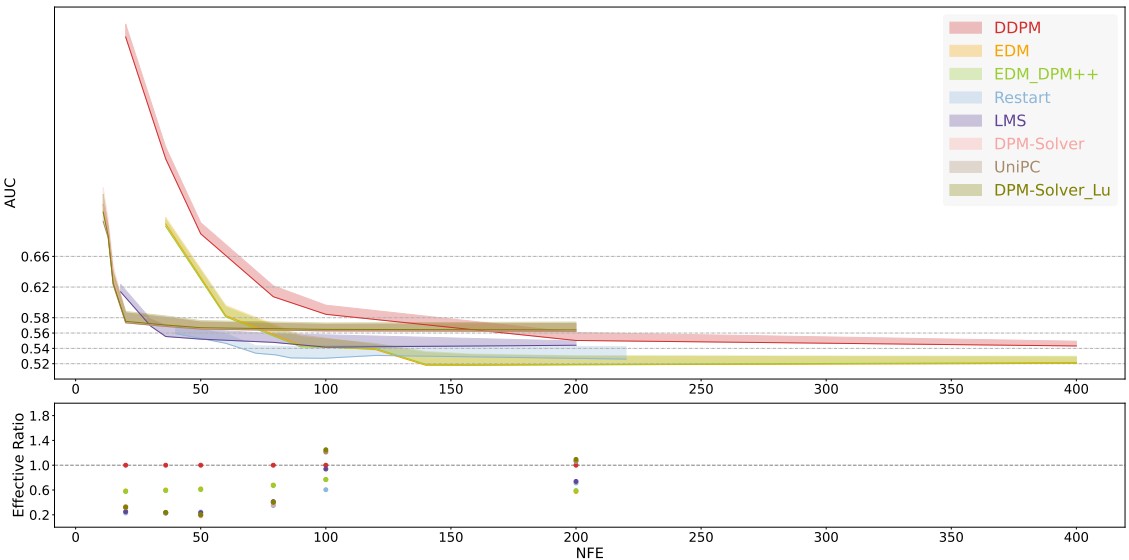

Figure 7: AUC values for different samplers as a function of evaluation steps, the shaded area is possible AUC value range for 20 evaluations through independent classifier training. The effective ratio is defined as the ratio between actual AUC from generated sample with different solvers minus the best possible AUC value 0.5, divided by the AUC value from generated sample with DDPM minus 0.5 $\frac{\text{AUC}_{gen}-0.5}{\text{AUC}_{DDPM}-0.5}$ Lower is better, value below 1 means better performance than DDPM at certain evaluation steps.

We showed the possible AUC values for high level feature classifiers with different samplers and schedulers by evaluating different samplers 20 times in Fig. 7. DPM-solver sampler seems to perform poorly on predicting the low and high-level features simultaneously. The Uni-PC sampler has very similar performance with DPM-solver sampler and only gives slightly better performance over very low sampling (10-20) steps. The lowest AUC is around 57 at 50th step. The EDM and Restart samplers surpass the previous 400 steps DDPM samplers at 50, 25 steps (100, 60 function evaluations) correspondingly, combined with other plots we showed, achieving the same and better performance with a 3-8 times speedup. The LMS ODE sampler has a better performance than EDM below 45 steps, but reaches a worse convergence than other samplers. The best AUC value for Restart samplers achieve at 30-36 and 79 steps, this might because we use the predefined configuration specifically for those steps. All proposed samplers have comparable and better performance with smaller steps. Most of the samplers we use actually have more parameters to tune than the baseline DDPM. For the common ODE samplers, a closer matching noise level during sampling would often yield better results. A possible future development is to fine-tune the scheduler configuration for ODE samplers in higher steps.

Now we take a look at one of the most difficult and important quantities for the low level voxel diffusion – $E_{\text{ratio}}$. In Fig.8, we choose 6 cases in our study. First, much faster convergences have been seen from new introduced samplers by 5-20 times. The obtained result matches with the Table.3 mentioned in Song's [59]. For EDM sampler, the separation power, defined as the triangular discriminator [8] $\frac{\sum_{i=1}^{n}\left(\text{count}_{\text{data},i}-\text{count}_{\text{gen},i}\right)^2}{\sum_{i=1}^{n}\left(\text{count}_{\text{data},i}+\text{count}_{\text{gen},i}\right)}$ which calculated from each bins of the histogram, quickly drops around 60 steps and achieves better performance than the baseline 400 steps DDPM sampler. The 20-30 steps DPM-Solver and UniPC samplers already have lower separation powers than 200 steps DDPM sampler. The 18 steps Restart sampler already could match the similar performance as the baseline. If we zoom-in the figure, at around 100-200

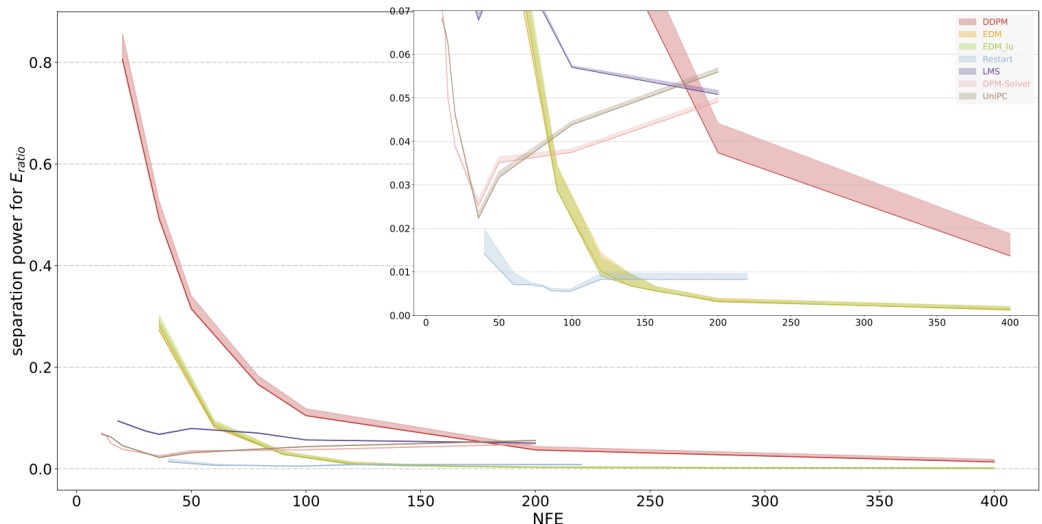

Figure 8: Separation power for $E_{\text{ratio}}$ with different samplers and schedulers as a function of evaluation steps, the shaded area is possible values of separation power for 20 independent samplings.

steps, EDM and Restart samplers have 3-5 times weaker separation power than the baseline DDPM samplers.

The preliminary Frechét Particle Distance (FPD) [58] values for some samplers are also provided in Table. 1. The FPD metric is relatively biased and unstable among subsamples we choose. So we follow the way from [23] to subtract the non-zero FPD value between two GEANT4 sample. Surprisingly, we find the EDM sampler converges even faster in this metric, surpassing the DDPM sampler around 36 steps (72 NFE). The 18-steps ($\sim$ 40 NFE) Restart sampler has slightly worse performance, this is mainly because we choose the predefined configuration for smallest $E_{\text{ratio}}$ separation power, a throughout fine-tune can make FPD value 20-30 percent smaller with almost same generation time. However, it still obtains the best FPD results with larger steps.

The separation power is decreased in smaller steps ($\leq 100$) but increased in larger steps by changing the EDM scheduler from Karras to Lu. The choice of schedulers provides a training-free method to balance the quality and speed of the backward diffusion process. The EDM/DPM-Solver combined sampler with Karras scheduler performs pretty similarly as the EDM sampler with Lu scheduler. The best performances of Restart samplers again are achieved in 30-36 steps. This sampler is crafted for efficiently conducting fast generation with outstanding performance and can be fine-tuned with post-training configuration. The DPM-solver exhibits a rapid drop in separation power within 10-20 steps but ceases to improve beyond those steps, achieving better performance than the 100-step DDPM around the 15[th] step. The plot further confirms that ODE sampler does better in low sampling steps as the discretization error of them are smaller than SDE ones in this region, the contraction of accumulated errors from SDE sampling process become much stronger than ODE so that a better convergence observed in higher sampling steps than ODE ones [48]. Some ODE samplers, such as LMS and DPM-solver, perform well on some metrics but not all. There is not a clearly superior option that can achieve the best quality while having the lowest generation time.

Table 1: Summary table for high-level feature classifier mean AUC, $E_{\text{ratio}}$ separation power, and Frechét Particle Distance [58] of of different samplers. Underline is the 400-step DDPM sampler (CaloDiffusion) as baseline, the $*$, $\dagger$, $\ddagger$, are the first, second, third best results. **Bold** is the highlighted result with either great performance or good quality-sampling time balance.

| Samplers(steps) | AUC ↓ | Separation power $(10^{-2})$ ↓ | FPD ↓ |
|---|---|---|---|
| DDPM(79) | 61.45±0.05 | 8.10±1.05 | 0.074±0.01 |
| DDPM(200) | 55.72±0.03 | 3.44±0.40 | 0.046±0.007 |
| DDPM(400) | 54.6±0.02 | 1.55±0.30 | 0.043±0.007 |
| EDM(36) | 55.3±0.06 | 2.56±0.72 | 0.035±0.007 |
| EDM(79) | **52.9±0.04** | 0.76±0.20 | **0.027±0.004** |
| EDM(200) | 52.6±0.03$^\dagger$ | 0.55±0.10$^\dagger$ | 0.023±0.004$^\ddagger$ |
| EDM(400) | 52.6±0.01$^*$ | **0.16±0.05$^*$** | 0.021±0.003$^*$ |
| EDM_DPM++(79) | 53.0±0.04 | 0.82±0.20 | 0.026±0.004 |
| EDM_Lu(79) | 52.8±0.04 | 0.86±0.20 | 0.026±0.004 |
| Restart(18) | 56.7±0.07 | 1.69±0.66 | 0.059±0.009 |
| Restart(36) | **54.0±0.06** | 0.73±0.26 | **0.025±0.003** |
| Restart(79) | 53.9±0.05$^\ddagger$ | 0.57±0.19$^\ddagger$ | 0.022±0.004$^\dagger$ |
| LMS(36) | 56.0±0.04 | 6.65±1.35 | 0.095±0.01 |
| DPM++(25) | 56.8±0.07 | 2.52±0.51 | 0.113±0.01 |
| UniPC(25) | 56.9±0.07 | **2.31±0.45** | 0.112±0.01 |

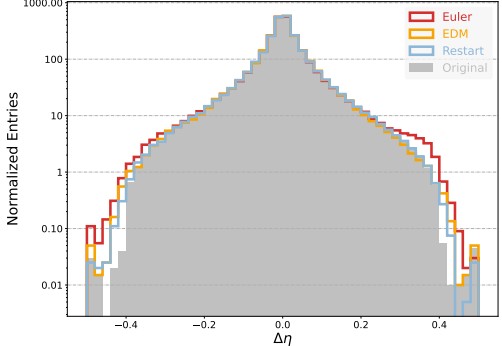
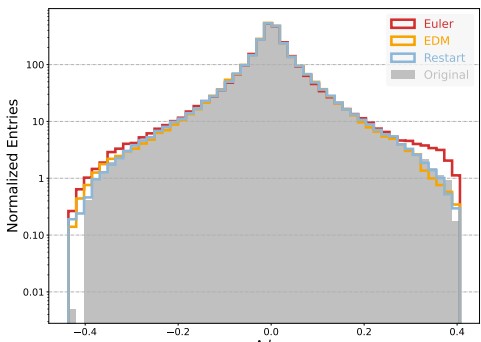

Figure 9: Comparison plots for relative angular coordinate of particle constituents inside gluon initiated jets with 200 steps DDIM Euler Sampler, 18 steps multi-level Restart Sampler, 50 steps EDM samplers. Left: $\Delta\eta$, right: $\Delta\phi$.

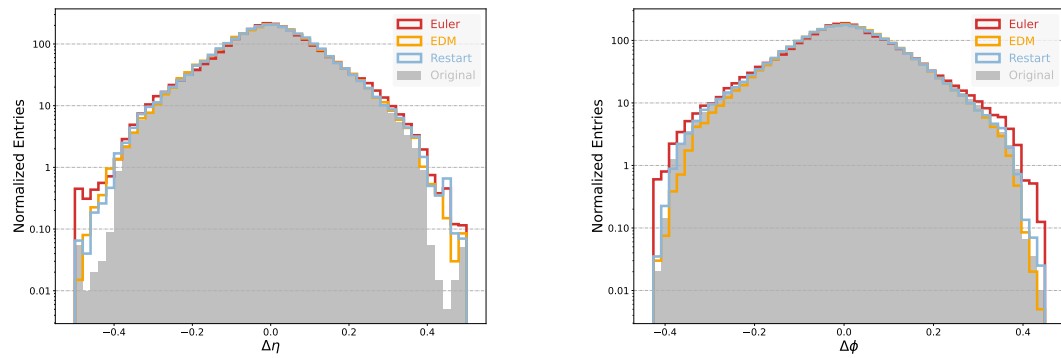

Figure 10: Comparison plots for relative angular coordinate of particle constituents inside top quark jets with 200 steps DDIM Euler Sampler, 18 steps multi-level Restart Sampler, 50 steps EDM samplers. Left: $\Delta\eta$, right: $\Delta\phi$.

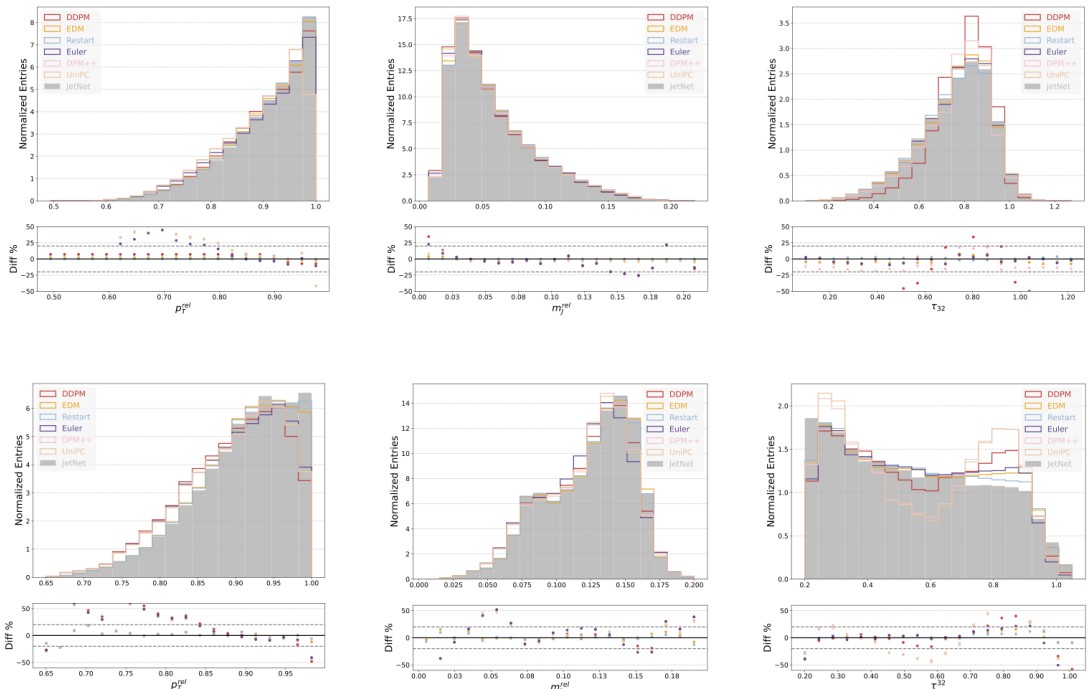

Figure 11: Comparison plots for substructure of particle constituents inside gluon/top quark jets with different samplers and schedulers.

Table 2: Summary table for low-level feature classifier mean AUC, Frechét Particle Distance calculated from jet-level classifier, and high-level features unbinned (Wasserstein) distance of different samplers on gluon-initiated jets top and gluon quark jets. Underline is the 400-step DDPM sampler (PC-JeDi) as baseline, the ⋆, †, ‡, are the first, second, third best results. **Bold** is the highlighted result with either great performance or good quality-sampling time balance.

| Samplers(NFE) | AUC ↓ | FPD ↓ | $W_m(10^{-4})$ | $W_{p_T}(10^{-4})$ | $W_{\tau_{32}}(10^{-3})$ ↓ |
|---|---|---|---|---|---|
| | | **Top** | | | |
| DDPM(200) | 52.8±0.04 | 0.16±0.03 | 13.54 | 12.04 | 16.05 |
| Euler(200) | 52.6±0.04 | 0.15±0.03 | 20.47 | 10.08 | 11.54 |
| EDM(100) | 51.9±0.03† | 0.07±0.01† | 7.65† | 7.07† | 8.32† |
| EDM_DPM++(100) | 52.0±0.02‡ | 0.08±0.005‡ | 8.05‡ | 7.17‡ | 8.34‡ |
| Restart(70) | **51.3±0.03**⋆ | **0.02±0.005**⋆ | 4.30⋆ | 5.65⋆ | 5.90⋆ |
| DPM++(25) | 53.7±0.05 | 0.20±0.04 | 25.64 | 10.45 | 25.76 |
| UniPC(25) | 54.0±0.07 | 0.22±0.03 | 27.89 | 10.49 | 26.04 |
| | | **Gluon** | | | |
| DDPM(200) | 52.5±0.04 | 0.10±0.03 | 5.69 | 6.01 | 14.30 |
| Euler(200) | 52.6±0.04 | 0.14±0.04 | 6.40 | 6.63 | 4.79† |
| EDM(100) | 51.9±0.03† | 0.07±0.01† | 5.03† | 4.65† | 4.92 |
| EDM_DPM++(100) | 52.0±0.03‡ | 0.08±0.01‡ | 5.40‡ | 4.59‡ | 5.03‡ |
| Restart(70) | **51.2±0.02**⋆ | **0.02±0.007**⋆ | **3.03**⋆ | 3.20⋆ | 3.15⋆ |
| DPM++(25) | 52.8±0.05 | **0.16±0.02** | 6.05 | 9.97 | 10.09 |
| UniPC(25) | 52.9±0.05 | 0.16±0.02 | 6.15 | 9.98 | 5.41 |

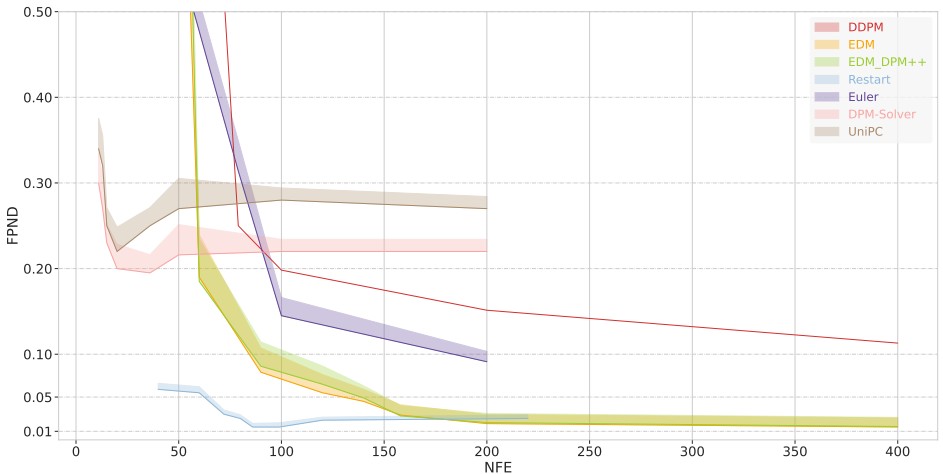

Figure 12: Comparison plots for FPD values for top jets as a function of evaluation steps with different samplers and schedulers. The shaded area is possible values of separation power for 10 independent samplings.

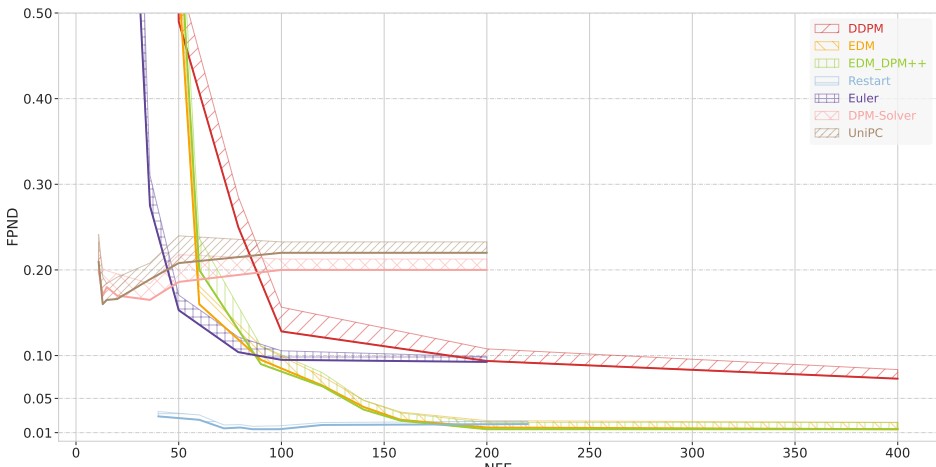

Figure 13: Comparison plots for FPD values for gluon jet as a function of evaluation steps with different samplers and schedulers. the shaded area is possible values of separation power for 10 independent samplings.

## 5.2 JetNet

We employ network training techniques with the same continuous noise level and representative samplers for the *JetNet* dataset to demonstrate that these methods are entirely portable across various datasets. As depicted in Fig. 9 and 10, the low level feature generations with a 50-steps EDM sampler and 18-steps Restart sampler show better performance than a 200-steps DDIM [36] Euler sampler on gluon jets. For more complicated topology, EDM and Restart have advantages in simulating the tails of the distribution over large $\eta$ region. EDM sampler has small deficiency for underestimating the particles from top quark jets after $\Delta\phi > 0.2$. This would have impact on reconstructing the jet level variables in Fig. 11.

The same training setup in PC-JeDi [25] is used with DDPM samplers as baseline. We found the Euler samplers could give a better performance with our new proposed training. Both EDM and Restart samplers capture most of the high level gluon jet features, with only few percent difference in relative transverse momentum and mass of jets. The improvement is clearer in more challenging substructure variables such as the n-subjetiness between subleading and third leading jets - $\tau_{32}$ when comparing with default DDPM and Restart samplers. Although some ODE samplers outperform than DDPM on $\tau_{32}$ of gluon jets, they can not achieve better overall performance, except LMS samplers. The largest discrepancy for this quantity appears around 0.7-0.9 region where the baseline DDPM sampler has around 30-40% difference. The low-steps Restart sampler gives obviously better generated distribution than others with less than 10% difference over most regions.

While simulating the complex distribution in a highly boosted environment, most of the samplers fail to achieve satisfactory alignment around the two asymmetric peaks in the relative mass distribution for top jets. We then anatomise the performance of proposed samplers via the three low level features with the same DNN classifier used in previous section, the values for the Fréchet Particle distance (FPD) calculated from the ParticleNet [60] multi-classifier with N = 6, and the unbinned metrics (Wasserstein distance) for reconstructed individual high level features. Performances are summarized on Table. 2. The Restart sampler provides the consistently best results for both top and gluon jets in all three metrics. Different scheduler variants on SDE samplers should not have too much influence on results. The 25-steps DPM-

Table 3: Summary table for Stochastic parameter and scheduler of EDM Heun Karras sampler.

| Dataset | $S_{\text{Churn}}$ | $S_{\max}$ | $S_{\min}$ | $S_{\text{noise}}$ | $\sigma_{\max}$ | $\sigma_{\min}$ |
|---------|-----------|-----------|-----------|-----------|-----------|-----------|
| *CaloCha.* | [15-40] | [20-40] | [0.01-0.1] | [1.001-1.004] | [40-80] | [0.002-0.01] |
| *JetNet* | [20-50] | [20-39] | [0.01-0.1] | [1.001-1.006] | [30-80] | [0.002-0.01] |

Solver sampler shows competitive result as the 200-steps traditional Euler and DDPM samplers on gluon jets.

In Fig. 12- 13, Restart sampler yields the best performance even in small number of function evaluations. Even though the boosts brought by those changes isn't as pronounced as they are in *CaloChallenge*, high step EDM and Restart samplers still have much lower FPD values than other samplers.

# 6 Conclusion and outlooks

In this study, we conducted a comprehensive investigation into the utilization of various state-of-the-art samplers and schedulers across diverse datasets in collider physics. We introduced a soft version of loss weighting techniques to expedite the convergence of physics-informed quantities towards a better reliable result. Our study presents several approaches for evaluating generated samples across different scenarios, providing a range of options for accelerating diffusion model without additional training or achieving superior results with fewer enhancements. Innovative sampling methods, such as Restart, exhibit better performance with just 18-30 steps, leading to a $\mathcal{O}(10)$ times speeding up without a significant performance loss. We showcase a potentially new baseline for training and sampling diffusion models in this rapidly evolving field.

Future development will aim to identify the optimal sets for these samplers on each dataset to achieve the maximum possible boost. Additionally, exploration into other areas, such as score distillation sampling and model fine-tuning, will be undertaken to obtain similar performance levels with even faster simulations.

## Acknowledgments

We thank Kevin Pedro and Oz Amram for fruitful discussions about details of `CaloDiffusion` project, and insightful comments for different evaluation metrics.

# A Hyperparameters of samplers and schedulers

We will list more details about the hyperparamter range we ended up choosing for each sampler.

For the Stochastic parameter we listed on Table. 3, the choices of $S_{\text{Churn}}$ and $S_{\min}$ are very important. In the *CaloChallenge* dataset, a too small $S_{\text{Churn}}$ would make the prediction worse over lower energy region, resulting a higher $E_{ratio}$ separation power, and a high $S_{\text{Churn}}$ would make noise injection power too high over low sampling step region to converge quickly. With our training setup, we usually set a larger $S_{\text{Churn}}$ on the *Jetnet* dataset. That is also one reason why a smaller improvement is observed here. The $S_{\text{noise}}$ does not affect too much on both dataset.

Table 4: Summary table for good parameter of Restart Karras sampler, 0 denoted as the single level Restart.

| **level** | $N_{\text{steps}}$ | $K$ | $\sigma_{\max}$ | $\sigma_{\min}$ |
|-----------|---------|--------|-----------------|-----------------|
| *0* | [2-5] | [2-15] | [1.09,0.30] | [0.14, 0.06] |
| *1* | [2-5] | [1-2] | [40.79] | [19.35] |
| *2* | [3-6] | [1-2] | [1.92] | [1.09] |
| *3* | [4-6] | [2-4] | [1.09,0.59,0.30] | [0.59,0.06] |
| *4* | [4-6] | [1-4] | [0.59,0.30] | [0.30,0.06] |
| *5* | [4-7] | [2-6] | [0.30] | [0.06] |

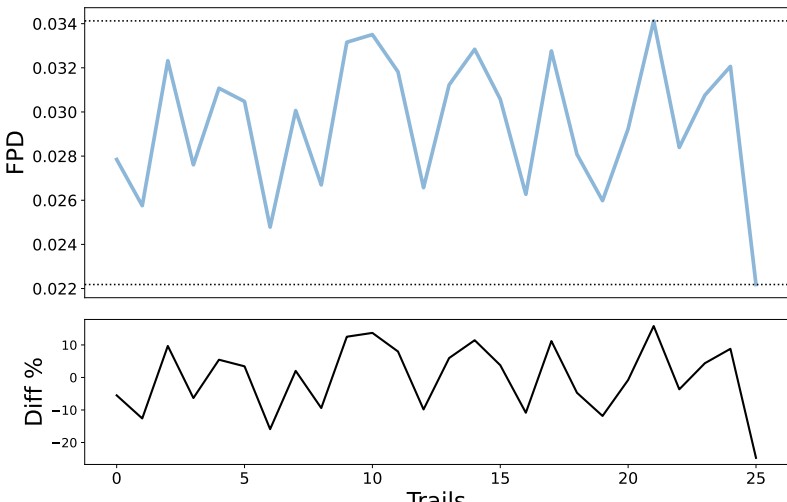

Figure 14: FPD values in CaloChallenge for 36 steps multilevel Restart samplers with 25 different configurations.

We also find several ways to constrain the multilevel configuration for generating satisfied results with Restart sampler by empirical evaluations.

Oftentimes different $N_{\text{steps}}$ across different level (range of time steps) yield better FPD value. Gradually increasing $N_{\text{steps}}$ toward last few steps give slightly better result than constant $N_{\text{steps}}$ among all levels. With a larger $K$ iteration value in 3$^{\text{rd}}$ and 5$^{\text{th}}$, we obtain the best results in current sampling steps, but also increase the generation time by few percent.

For DPM-solver, we choose a slightly different noise level. The $\beta_{\max}, \beta_{\min}$ for VP scheduler is $19.9, 0.002$. For Karras and Lu scheduler, $\sigma_{\max}, \sigma_{\min}$ is $[15, 40], [0.001, 0.01]$. For LMS, we choose $[3, 4, 6]^{\text{th}}$ order of calculation. The UniPC sampler is using 2$^{\text{nd}}$ order $B(h)$-2 solver with the same scheduler as DPM-solver.

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
