# Peer review of "Choose Your Diffusion: Efficient and flexible ways to accelerate the diffusion model in fast high energy physics simulation"

_SciPost Physics, doi:SciPost Phys. 18, 195 (2025)_

## Round 1 · Referee Report · Anonymous (Referee 1) · 2024-5-1

Strengths
1: Consideration of multiple solvers for diffusion generative models
Weaknesses
1: The text would greatly benefit from a revision. The discussions of the results are hard to follow. 2: The authors aim to have a comprehensive comparison of samplers, but fall short on being through with their studies. How different distributions change with different sampling steps? How their results compare against public results on the same dataset? 3: Even though the JetNet dataset is mentioned, the studies performed using the dataset are considerably limited compared to the previous sections, lacking any conclusive results.
Report
The authors aim to provide a comprehensive comparison between samplers for diffusion generative models. They use 2 public datasets to evaluate the differences between samplers. The calochallenge dataset 2 and the JetNet dataset. While the choice of samplers covers a comprehensive number of modern and traditionally used samplers, the results are hard to follow. In particular, in almost no plot all samplers are shown simultaneously, with only an arbitrary subset chosen for each distribution. This issue is more acute in the JetNet dataset where almost no effort is used to quantify the differences between solvers. Additionally, the authors do not compare their results with public benchmarks, undermining their goal of establishing the choice of better solvers for specific tasks. The goal of comparing multiple solvers is indeed interesting and deserves to be published, but in the current form additional studies and textual improvements are necessary. More detailed feedback is given below.
P2: “As a consequence, the detector simulations for the modern high granularity detectors occupy the most computation resources. ” Citations to support this statement would be great.
P3: “The performance surpasses the previous model by several evaluation metrics.” Where is the evaluation metrics?
P5: “ODE solvers often have smaller discretization errors than SDE solvers in smaller sampling steps.” Is there a reference to support this statement? ODE solvers often require less steps than SDE solvers, which contradicts the argument of bigger time steps given.
P5: VP scheduler: the variance preserving property is determined by the relationship alpha^2 + sigma^2 = 1 and not by the time evolution of sigma. I would point to it when introducing to the VP schedule or changing the name to avoid confusion (for example, cosine is also a VP schedule).
P5: EDM schedule:
- What does churn means in this context? the meaning of S_{churn} is given but would be nicer to define an acronym related the meaning of the acronym.
- “This SDE sampler, unlike other SDEs that only give satisfied result after a very large sampling steps, can converge quickly because of this” satisfying instead of satisfied. What does “this” refer to in the explanation?
Eq8:
- How is the score function of the data x dependent on time? This result is only true for sigma = t, which is only one of the possible schedulers discussed. How is that implemented for the other schedulers?
- What is D?
Eq:9
- How Eq.8 gives you Eq. 9? What if F?
- Define the noise epsilon and the relationship with the noise applied to the data.
P8: “To mitigate the instability of the optimization over the training, …” why is the training unstable to begin with? How is SNR(t) defined?
Eq. 10: What is the benefit of the function being even? Are negative time values used at any point?
P9: “In addition, a DNN classifier with 2 layers, each comprising 2048 nodes and a 0.2 dropout rate” how is that determined to be sufficient?
Fig. 2: The authors claim the benefit of the weight function based on the convergence results as a function of the number of training epochs. This argument is unfortunately not sufficient to prove their point as the energy ratio is but a single observable from a high dimensional dataset. Moreover, additional parameters such as the choice of optimizer, learning rate, and batch size will all influence the convergence rate independently from the choice of weighting scheme. Additionally, faster training convergence is a debatable quantity for a generative model, as the main benefit of fast detector simulation is at inference time, with training time corresponding to a small fraction compared to the expected inference time during production.
Fig. 3: The ratio plots should be zoomed in as currently the axis range is too big compared to the plot. The choice of plots is also odd as other schedulers were also discussed in the previous sections. The same plot with all schedulers shown at the same time with zoomed in axis in the ratio plot would be better to compare the differences in generation quality.
Similarly, the number of steps chosen for each scheduler seems arbitrary at this point. How were they chosen?
Figs 3, 4, 5: Again, even though multiple solvers are described, the authors only show results for an arbitrary subset. Either show the results for all samplers, or motivate why the EDM is preferred in these plots.
Fig. 6: Why the ratio is not shown? This is the first distribution showing a bigger set of schedulers and would benefit from the ratio plot. Why EDM is shown with different number of steps? Would the other samplers also improve with more steps? For example, LMS shows a disagreement at low voxel energies, but uses only 36 steps. Similarly to my previous question, the authors should motivate how the choice of steps shown in the comparison plots are motivated, otherwise differences cannot be attributed to the solvers but simply from a poor choice of number of steps.
P10: “Indeed some of them are struggling to match the low voxel energy, the presence of the tail is probably a consequence of model itself and too low energy threshold”. What does that mean? That the model itself is not good enough? If so, then no sampler should be able to get a good agreement in the low energy voxel region, which is not true from Fig. 6.
P11: “LMS sampler involves an additional parameter "order" of the coefficient which makes the generation time longer as it increases”. This sentence is very cryptic as that parameter has not been introduced nor is it explained how it influences anything in the solver.
Fig. 7: Why LMS seems to increase instead of decrease with more steps? This plot and results would be great to show early in the text to motivate the choice of sampling steps picked for individual histograms (if that is true that the number of steps were chosen based on this plot).
Similarly, plots showing, as a function of the number of steps and or each sampler, distributions such as the chi-square or EMD for the 1-dimensional histograms shown before would be a great way to compare the samplers.
P12: “First, much faster convergences have been seen from all new introduced samplers” in the context of this paper, all samplers are new. Please be more specific about the samplers referred to.
Fig. 8: How is separation power defined?
Fig. 9: Again, a ratio plot would be beneficial to aid the discussions on the differences observed between samplers. How many steps is high EMD steps?
P12: “This is crucial for us to perform accurate energy calibration from low-level fast calorimeter simulation later.” I’m missing how the previous discussion reaches this conclusion.
Table 1: What bold entries mean? The best results? In the AUC column, there are lower AUC and FPD values than the ones shown in bold. Uncertainties from multiple runs should also be shown for each metric to identify when differences are actually significant.
P14: “We choose Karras and Lu schedulers to illustrate the impacts of different noise schedulers on the same samplers.” Why this choice of samplers? Where is this illustrated? The following discussion is very hard to follow without any visual aid.
P16: The jetnet results are incredibly short compared to the calorimeter results. How the sampling quality changes in this case versus the number of steps used? How the values you obtain compare with the many public results on the jetnet dataset?
“It may be because methods are more applicable to UNet and pixelated data than point clouds network.” Why would it be the case? Which studies were performed to reach this conclusion?
Recommendation
Ask for major revision
Warnings issued while processing user-supplied markup:
- Inconsistency: Markdown and reStructuredText syntaxes are mixed. Markdown will be used.
Add "#coerce:reST" or "#coerce:plain" as the first line of your text to force reStructuredText or no markup.
You may also contact the helpdesk if the formatting is incorrect and you are unable to edit your text.
General response:
Thank you very much for the insightful suggestions! we have updated most of the plots on the script. Now a clearer view for performance of all samplers with different variants. Specifically on Fig.5,6,7,8,12. Table.2 The evaluations on JetNet dataset is extensively studied compared with previous script. We also provide additional explanation on result section.5. We do compare with the benchmark methods. The performance metric values showed in CaloChallenge is from pretrained CaloDiffusion model with DDPM sampler, in JetNet is from PC-Jedi with Euler/DDPM sampler, which both are currently one of the best diffusion models in event/detector simulations we have in the field We declared used the pretrained model in the text and comparison plot.
P2: “As a consequence, the detector simulations for the modern high granularity detectors occupy the most computation resources. ” Citations to support this statement would be great.
A: The reference for general challenges traditional full simulation would face in HL-LHC has been cited in the introduction, thanks for the comments
P3: “The performance surpasses the previous model by several evaluation metrics.” Where is the evaluation metrics?
A: We have cited the previous papers which provide few benchmark metrics to this study (earlier one from DCTRGAN: 2009.03796, and recent one 2410.21611) , most of the evaluation metrics are publicly available.
P5: “ODE solvers often have smaller discretization errors than SDE solvers in smaller sampling steps.” Is there a reference to support this statement? ODE solvers often require less steps than SDE solvers, which contradicts the argument of bigger time steps given.
A: There are bunch of reference studies from diffusion model in Computer Vision area which provide some theoretical formulation and empirical studies to support the statement, most relevant ones are from DPM-solver(2211.01095), EDM (2206.00364), and Restart paper(2306.14878), we have cited them in the text.
P5: VP scheduler: the variance preserving property is determined by the relationship alpha^2 + sigma^2 = 1 and not by the time evolution of sigma. I would point to it when introducing to the VP schedule or changing the name to avoid confusion (for example, cosine is also a VP schedule).
A: Apologize for the confusion, we add some more explanation on the samplers and schedulers with finer math details. Variance Preserving schedulers as a more generalized version of cosine beta schedulers used in both score matching and DDPM, defined as a continous time step where $\beta_\mathrm{min}$ and $\beta_\mathrm{max}$ are constant to determine the lower and upper bound of noise level.
P5: EDM schedule: - What does churn means in this context? the meaning of S_{churn} is given but would be nicer to define an acronym related the meaning of the acronym. - “This SDE sampler, unlike other SDEs that only give satisfied result after a very large sampling steps, can converge quickly because of this” satisfying instead of satisfied. What does “this” refer to in the explanation?
A (to above two questions): The text has been changed. This is referring to the stochastic parameters tuning, which DDPM don’t have too many post-training parameters to be tuned.
Eq8: - How is the score function of the data x dependent on time? This result is only true for sigma = t, which is only one of the possible schedulers discussed. How is that implemented for the other schedulers? - What is D?
Eq:9 - How Eq.8 gives you Eq. 9? What if F? - Define the noise epsilon and the relationship with the noise applied to the data.
A (to above three questions): Thanks for the clarification. In the context, we unify the timestep definition, the finer math formulation is added to help understanding the content
P8: “To mitigate the instability of the optimization over the training, …” why is the training unstable to begin with? How is SNR(t) defined?
A: This is really a good question! the training for diffusion is hard to converge, or it's hard to tell the training or sampling outcomes just from the convergence of loss, as the forward part is to make the target toward some high-dim gaussian, there might be data with "high frequency" which hardly can see a loss drop over numbers of epochs, those SNR weighting methods are used commonly in some paper up to now. https://arxiv.org/abs/2310.14189 https://arxiv.org/pdf/2406.14548
Eq. 10: What is the benefit of the function being even? Are negative time values used at any point?
A: The way to design this weight is to be more manageable to tune instead of just fixing a value which may hardly cover whole noise level region. Given the square on both denominator and numerator, the suggested positive constant sigma_c value, there is no clear advantages for function being even as we won’t considered negative time values.
P9: “In addition, a DNN classifier with 2 layers, each comprising 2048 nodes and a 0.2 dropout rate” how is that determined to be sufficient?
A: It is the default public metrics used for most of fast calorimeter simulation models, in the future, we wish to extend the evaluation metrics to be more robust, using a ResNet or heavier model to get the full test.
Fig. 2: The authors claim the benefit of the weight function based on the convergence results as a function of the number of training epochs. This argument is unfortunately not sufficient to prove their point as the energy ratio is but a single observable from a high dimensional dataset. Moreover, additional parameters such as the choice of optimizer, learning rate, and batch size will all influence the convergence rate independently from the choice of weighting scheme. Additionally, faster training convergence is a debatable quantity for a generative model, as the main benefit of fast detector simulation is at inference time, with training time corresponding to a small fraction compared to the expected inference time during production.
A: Thank you for sharing your thoughts. We have observed that the use of loss weighting techniques is frequently highlighted in recent diffusion model papers. These techniques serve as an effective way to enhance network performance without introducing additional computational costs during training or inference. While faster training may not be the primary advantage of generative models, it can still be beneficial for achieving a more manageable and efficient model development process. The advantages for using a noise level dependent weight seen by not only one single training, by fixing all the training hyper parameters the same while only changing our weighting scheme
Fig. 3: The ratio plots should be zoomed in as currently the axis range is too big compared to the plot. The choice of plots is also odd as other schedulers were also discussed in the previous sections. The same plot with all schedulers shown at the same time with zoomed in axis in the ratio plot would be better to compare the differences in generation quality. Similarly, the number of steps chosen for each scheduler seems arbitrary at this point. How were they chosen? Figs 3, 4, 5: Again, even though multiple solvers are described, the authors only show results for an arbitrary subset. Either show the results for all samplers, or motivate why the EDM is preferred in these plots.
A: We have updated some plots and tables to include results for all discussed samplers. Additionally, we have provided a rationale for why EDM and Restart is preferred in certain contexts within the text, emphasizing its superior performance in specific scenarios. Choice of Number of Steps: The number of steps for each scheduler was chosen to demonstrate how our new samplers perform with low sampling steps. Our goal was to illustrate that both the EDM and restart samplers achieve high performance with fewer steps, the efficiency and benefits of using these samplers. We have added a detailed explanation of this rationale in the revised manuscript to clarify our choice. While we initially focused on a subset of solvers to highlight key findings, presenting a comprehensive view is also very important.
Fig. 6: Why the ratio is not shown? This is the first distribution showing a bigger set of schedulers and would benefit from the ratio plot. Why EDM is shown with different number of steps? Would the other samplers also improve with more steps? For example, LMS shows a disagreement at low voxel energies, but uses only 36 steps. Similarly to my previous question, the authors should motivate how the choice of steps shown in the comparison plots are motivated, otherwise differences cannot be attributed to the solvers but simply from a poor choice of number of steps.
A: The ratio plots are added and arrows to indicate when the samplers start to perform poorly. The choice of the number of steps for each scheduler was based on the need to balance performance and inference time. In many practical applications, it is not feasible to use a very high number of steps (e.g., 200-400 steps) due to time constraints. Our goal was to demonstrate that good performance can be achieved with a relatively low number of steps. This decision was made to reflect realistic scenarios where inference time is a critical factor. The figures and tables are also updated with a comprehensive overview for many samplers with different sampling steps, to anatomize those methods in another perspective
P10: “Indeed some of them are struggling to match the low voxel energy, the presence of the tail is probably a consequence of model itself and too low energy threshold”. What does that mean? That the model itself is not good enough? If so, then no sampler should be able to get a good agreement in the low energy voxel region, which is not true from Fig. 6.
A: The statement "Indeed some of them are struggling to match the low voxel energy; the presence of the tail is probably a consequence of the model itself and too low energy threshold" was intended to highlight that certain samplers have difficulty achieving accuracy in the low voxel energy regions. This difficulty is not necessarily due to the inadequacy of the model but rather the sensitivity of the sampling process to low energy thresholds. Our model is generally capable of producing accurate results across various energy levels, as demonstrated by the performance of some samplers in Fig. 6. There is indeed a difficulty that we find no samplers can capture all the voxel regions well at the same time, either for intermediate regions or low energy regions.
P11: “LMS sampler involves an additional parameter "order" of the coefficient which makes the generation time longer as it increases”. This sentence is very cryptic as that parameter has not been introduced nor is it explained how it influences anything in the solver.
A: We have rewritten the sentences.
Fig. 7: Why LMS seems to increase instead of decrease with more steps? This plot and results would be great to show early in the text to motivate the choice of sampling steps picked for individual histograms (if that is true that the number of steps were chosen based on this plot).
A: Some of the ODE samplers would give good results at small steps, at larher steps as the total error accumulated to larger, it tend to “overfit” a bit, we choosed the reasonable steps based on this plot to draw the individual histograms
Similarly, plots showing, as a function of the number of steps and or each sampler, distributions such as the chi-square or EMD for the 1-dimensional histograms shown before would be a great way to compare the samplers.
A: we changed number of steps to number of function evaluations (evaluations), we have Eratio separation power works similar with chi2, we found this quantity is hard to study well without any extra post processing normalisation, whereas other quantities as we showed in Fig 4-6 can’t really tell the quantity of samplers. Oftentimes the general qualities of the samplers are very relevant to those metrics we choose like high-level AUC, ERatio separation power and FPD.
P12: “First, much faster convergences have been seen from all new introduced samplers” in the context of this paper, all samplers are new. Please be more specific about the samplers referred to.
A: This particularly seen from EDM,Restart and few ODE samplers from the beginning
Fig. 8: How is separation power defined?
A: We have defined the metrics as the same they used in previous paper, the triangular discriminator in each bins.
Fig. 9: Again, a ratio plot would be beneficial to aid the discussions on the differences observed between samplers. How many steps is high EMD steps?
A: We have updated the figures with ratio plots, high EMD refers to 79 steps
P12: “This is crucial for us to perform accurate energy calibration from low-level fast calorimeter simulation later.” I’m missing how the previous discussion reaches this conclusion.
A: We have removed some of the redundant texts to make the result section easier understood.
Table 1: What bold entries mean? The best results? In the AUC column, there are lower AUC and FPD values than the ones shown in bold. Uncertainties from multiple runs should also be shown for each metric to identify when differences are actually significant.
A: The bold entries mean samplers with top quality/speed tradeoff, for some SDE samplers, the sampling quality could be good with many steps, but can’t use them often in realistic case as limited GPU budget, we updated the tables with all the samplers with some step choices and uncertainties included.
P14: “We choose Karras and Lu schedulers to illustrate the impacts of different noise schedulers on the same samplers.” Why this choice of samplers? Where is this illustrated? The following discussion is very hard to follow without any visual aid.
A: Those are the currently mainstream schedulers used in continuous-time DM, we choose them to see if a simple post-training change could yield better results.
P16: The jetnet results are incredibly short compared to the calorimeter results. How the sampling quality changes in this case versus the number of steps used? How the values you obtain compare with the many public results on the jetnet dataset? A: We have added many JetNet evaluation metrics, especially on the table for samplers we introduced
“It may be because methods are more applicable to UNet and pixelated data than point clouds network.” Why would it be the case? Which studies were performed to reach this conclusion?
A: We have removed some of the redundant texts to make the result section easier understood.

Author: Sitian Qian on 2025-04-15 [id 5371]
(in reply to Report 3 on 2024-05-08)Warnings issued while processing user-supplied markup:
Add "#coerce:reST" or "#coerce:plain" as the first line of your text to force reStructuredText or no markup.
You may also contact the helpdesk if the formatting is incorrect and you are unable to edit your text.
Major:
The paragraph starting “However,” on page 12 is difficult to follow. Some claims in the text are not substantiated in the figures because the relevant method is missing. For example, the Uni-PC sampler does not appear in Figs 2-9. I also do not understand what is meant by “The EDM and Restart samplers surpasses the previous 400 steps DDPM samplers at 50, 25 steps correspondingly” since in Figure 7, the DDPM sampler appears to be the worst everywhere.
Page 14: “The best performances of Restart samplers again are achieved in 30-36 steps.” This does not appear to be true based Table 1, since Restart(79) has better metrics. Uncertainties should be included in the table.
Why does Table 2 contain such a small subset of the solvers in Table 1?
Minor: - Figure 1 contains elements that are not explained in the caption, making the diagram difficult to interpret.
Page 3: Equation (3) seems redundant.
Page 5: For completeness, the first sampler in the list should be described.
Page 8: The relationship between Eqs. 8 and 9 is not made explicit. In particular, Fθ is not defined.
Page 9: The separation power should be defined.
Attachment:
wrapper_SciPostPhysLectNotes_diff_compressed_MKIPHju.pdf

---

## Round 1 · Referee Report · Anonymous (Referee 2) · 2024-5-7

Strengths
-
The paper provides a comprehensive overview of the impact of different sampling algorithms for diffusion based generative models. The authors assess how these choices impact sample quality and generation time.
-
The authors show significant speedups in generation time with good quality as compared to the baseline sampling methods used in prior works
Weaknesses
1.The results from the many samplers tried are not always presented in a clear and organized way, making it overly difficult to draw conclusions.
-
The discussion and results shown on the JetNet dataset are very limited as compared to the results shown on the CaloChallenge dataset
-
The paper is mostly building on and improving prior work and is not very innovative.
Report
Requested changes
Major comments:
-
The results Sec. 5.1 on pgs 10-13 should be presented in a more organized and clear way. Many different samplers & noise schedules are defined in the introduction, but results are shown for only a subset of them without much organization. Furthermore, different combinations are compared when discussing different shower features.The discussion in the text therefore quite difficult to follow and it is hard to gain insight into what the key takeways of these comparisons are. The text on pg 14 is more clear.
-
Comparisons are given in terms of number of sampling steps. However, the metric one will care about in practice is sample generation time, which is directly related to the number of denoising model evaluations required to produce a sample. Some of the samplers used are first order, requiring only one evaluation of the denoising model per step, while others are higher order, requiring multiple model evaluations per step. It would be better to present results (eg Fig. 7, table 1) in terms of number of model evaluations rather than sampling steps. This would give the reader a better understanding of the tradeoff between quality vs generation time of each sampling method
-
The discussion of the results on the JetNet dataset is very limited and does not seem to add much to the paper. It seems much fewer combinations of samplers are tried with no justification given as to why. Since all sampling methods are seen to given good performance it would be interesting to try to push towards a lower sampling step regime where there might be more pronounced differences. Plots similar to Fig. 7 and 8 would be informative.
-
The authors often provide reasonable ranges for hyper parameters which is useful for building intuition, but the exact hyper parameter choices of the authors are not given. For reproducibility purposes, for any results shown in a plot / table the exact hyperparameters used should be specified somewhere in the text (an appendix is fine)
Minor comments:
- There are many grammatical mistakes or awkward phrasing in the text of the paper. Some of these are innocuous and do not harm the clarity of the ideas, but other times it is difficult to understand the authors meaning. Some examples of sentences I found difficult to understand are:
"This scheduler is even more decreasing than Karras schedulers, so the step size changes more rapidly in low sampling step while more slowly in last steps."
"Because the Restart samplers works with a predefined parameters for configuring the EDM ODE. The generation time is directly proportional with the restart iterations in the configuration, usually longer than default EDM samplers."
"All proposed samplers have comparable and better performance with smaller steps. Indeed some of them are struggling to match the low voxel energy, the presence of the tail is probably a consequence of model itself and too low energy threshold"
-
When describing the EDM sampler "In our study, such degradation only happens when extreme values are selected." Please clarify what 'extreme' values means in this case
-
Description of the Lu (last bullet on pg 5) scheduler is not very clear, particularly the last bullet
-
Reference should be given for all samplers on Pg 6, currently not all are properly referenced
-
It appears that the JetNet subsection (4.2) covers all the content on pg 8 but it actually only refers to the first three lines. A new subsection (4.3) should be started to better differentiate this
-
Fig. 3 caption does seem correct. The caption says it is center of energy but what is being shown is the average energy as a function of the radial bin.
-
In general plots should have fully descriptive labels on the plots themselves rather than relying on the figure caption to distinguish otherwise identical labels. Eg in Fig. 2 each plot should have a unique label telling the number of epochs and whether the default or min-SNR reweighting is used.
Recommendation
Ask for major revision
Warnings issued while processing user-supplied markup:
- Inconsistency: Markdown and reStructuredText syntaxes are mixed. Markdown will be used.
Add "#coerce:reST" or "#coerce:plain" as the first line of your text to force reStructuredText or no markup.
You may also contact the helpdesk if the formatting is incorrect and you are unable to edit your text.
Major comments:
- The results Sec. 5.1 on pgs 10-13 should be presented in a more organized and clear way. Many different samplers & noise schedules are defined in the introduction, but results are shown for only a subset of them without much organization. Furthermore, different combinations are compared when discussing different shower features. The discussion in the text therefore quite difficult to follow and it is hard to gain insight into what the key takeways of these comparisons are. The text on pg 14 is more clear.
A: Thank you for the good comments! We have updated some plots and tables to include results for all the samplers in the discussion, also modify some wording in the result section to make the context easier to follow. The comparison should be more comprehensive this time. And the point we really want to claim here is: by only changing part of the diffusion model (i.e. some training-free sampling process), the generation cost could be either reduced, or the sample quality can be improved with a much better computational cost tradeoff.
- Comparisons are given in terms of number of sampling steps. However, the metric one will care about in practice is sample generation time, which is directly related to the number of denoising model evaluations required to produce a sample. Some of the samplers used are first order, requiring only one evaluation of the denoising model per step, while others are higher order, requiring multiple model evaluations per step. It would be better to present results (eg Fig. 7, table 1) in terms of number of model evaluations rather than sampling steps. This would give the reader a better understanding of the tradeoff between quality vs generation time of each sampling method
A: That’s a really good comment! Indeed some of the samplers we used are first order (Euler, DPM-Solver), but others like EDM, found to be better in Heun (2rd order step), and samplers like LMS increase the sampling times by larger “order” hyperparameter, the the actual sampling steps of Restart samplers depends on the number of iterations on certain time interval. We unify the metrics for all of the samplers on a number of evaluation steps.
- The discussion of the results on the JetNet dataset is very limited and does not seem to add much to the paper. It seems much fewer combinations of samplers are tried with no justification given as to why. Since all sampling methods are seen to given good performance it would be interesting to try to push towards a lower sampling step regime where there might be more pronounced differences. Plots similar to Fig. 7 and 8 would be informative.
A: Thanks for the insightful suggestions. We have added details for JetNet results with many combinations for all samplers proposed. We also pretty agreed with a focus on lower sampling step or NFE regime as the improvement on metrics is more clear there. We have updated those plots for CaloChallenge, and also added the FPD against a wide range of NFE for different samplers for JetNet. The clear advantages can be seen from Restart and EDM samplers.
- The authors often provide reasonable ranges for hyper parameters which is useful for building intuition, but the exact hyper parameter choices of the authors are not given. For reproducibility purposes, for any results shown in a plot / table the exact hyperparameters used should be specified somewhere in the text (an appendix is fine)
A: We have updated the hyperparameters in appendix
Minor comments:
- There are many grammatical mistakes or awkward phrasing in the text of the paper. Some of these are innocuous and do not harm the clarity of the ideas, but other times it is difficult to understand the authors meaning. Some examples of sentences I found difficult to understand are:
"This scheduler is even more decreasing than Karras schedulers, so the step size changes more rapidly in low sampling step while more slowly in last steps."
"Because the Restart samplers works with a predefined parameters for configuring the EDM ODE. The generation time is directly proportional with the restart iterations in the configuration, usually longer than default EDM
samplers."
"All proposed samplers have comparable and better performance with smaller steps. Indeed some of them are struggling to match the low voxel energy, the presence of the tail is probably a consequence of model itself and too low energy threshold"
A: There are some non-native expressions with grammar mistakes, the content for the result section has been modified carefully.
- When describing the EDM sampler "In our study, such degradation only happens when extreme values are selected." Please clarify what 'extreme' values means in this case
A: Thank you for spotting this! The extreme could refer to large gamma (noise on each intermediate timestep, or S_churn to start adding noise), bad choice of those SDE parameter may affect the results
- Description of the Lu (last bullet on pg 5) scheduler is not very clear, particularly the last bullet
- Reference should be given for all samplers on Pg 6, currently not all are properly referenced
A: The relevant citation has been added
- It appears that the JetNet subsection (4.2) covers all the content on pg 8 but it actually only refers to the first three lines. A new subsection (4.3) should be started to better differentiate this
A: We did the minor modification on paper structure
- Fig. 3 caption does seem correct. The caption says it is center of energy but what is being shown is the average energy as a function of the radial bin.
- In general plots should have fully descriptive labels on the plots themselves rather than relying on the figure caption to distinguish otherwise identical labels. Eg in Fig. 2 each plot should have a unique label telling the number of epochs and whether the default or min-SNR reweighting is used.
A: We have updated the plots and correct the text
Attachment:

---

## Round 1 · Referee Report · Anonymous (Referee 3) · 2024-5-8

Strengths
1- The range of techniques explored in the study is wide 2- The results indicate that a significant reduction in generation time for diffusion models is possible without additional training
Weaknesses
1- The descriptions of each technique are brief. 2- Most plots lack error bars, undermining the qualitative comparisons drawn from them. 3- Some claims are not clearly reflected in the results.
Report
On the other hand, the techniques that aim to improve the convergence during training are not thoroughly studied. Some statements are not clearly supported by the results presented, often because the relevant result is missing from a figure. In general, the clarity of the manuscript should be improved prior to publication.
Requested changes
Major: - In section 2, the loss functions for DDPM and SGM are not given. The authors write “The noise scheduler determines how fast those samplers should learn” but this can only be understood in terms of a loss function. The first loss function in the paper appears in Eq. 9, but it is placed in the JetNet section, which does not seem appropriate.
-
The third and fourth schedules in the list on page 5 are written as $t_{i<N} = ...$ , but this is inconsistent with the notation used previously.
-
Regarding Figure 2:
- In the first line on page 10, the authors refer to “default EDM training loss weight”, but as far as I can tell, this has not been introduced. Does this refer to the min-SNR weight proposed in [51]? If not, then the Figure is also lacking a comparison between the original minSNR weighting and the one given in Eq. 10.
- Can the authors explain why the minSNR-trained model is so poor at 30 epochs?
-
I find Figure 3 to be uninformative, given that there is such a small difference between the plots and no uncertainties are shown. In fact, I question the conclusion that EDM improves with more steps, since it appears as though the 30-step EDM is better.
-
In Figures 7,8 (and throughout the paper) it is not made clear whether ‘steps’ refers to the total number of function evaluations or just to the number of integration steps. If the latter is true, then the figures are misleading since a given step would not equate to a fixed sampling time. In this case, I would suggest updating the figures to use number of function evaluations.
-
In Figure 5, can the authors explain why the agreement for $E_\mathrm{total}$ is so good while $E_\mathrm{total} / E_\mathrm{inc}$ is poor in Figure 2 ? Given that $E_\mathrm{inc}$ is a constant, I would expect both plots to indicate similar performance. I also note that this figure is not discussed in the text.
-
The paragraph starting “However,” on page 12 is difficult to follow. Some claims in the text are not substantiated in the figures because the relevant method is missing. For example, the Uni-PC sampler does not appear in Figs 2-9. I also do not understand what is meant by “The EDM and Restart samplers surpasses the previous 400 steps DDPM samplers at 50, 25 steps correspondingly” since in Figure 7, the DDPM sampler appears to be the worst everywhere.
-
Page 14: “The best performances of Restart samplers again are achieved in 30-36 steps.” This does not appear to be true based Table 1, since Restart(79) has better metrics. Uncertainties should be included in the table.
-
Why does Table 2 contain such a small subset of the solvers in Table 1?
Minor: - Figure 1 contains elements that are not explained in the caption, making the diagram difficult to interpret.
-
Page 3: Equation (3) seems redundant.
-
Page 5: For completeness, the first sampler in the list should be described.
-
Page 6: In the EDM description, the stochasticity is not apparent: from which distribution is $S_{churn}$ sampled?
-
Page 8: The relationship between Eqs. 8 and 9 is not made explicit. In particular, $F_\theta$ is not defined.
-
Page 9: The separation power should be defined.
-
The authors sometimes describe the samplers as ‘learning’ features, but I find this misleading since they are only relevant after the neural network is trained.
Recommendation
Ask for major revision

---

## Round 2 · Referee Report · Anonymous (Referee 1) · 2025-4-16

Report

The authors carefully considered all the criticism points raised in the first iteration of the review and greatly improved the results obtained in their research. I'm happy to support the publication of this work as is.

Recommendation

Publish (meets expectations and criteria for this Journal)

---

## Round 2 · Referee Report · Anonymous (Referee 2) · 2025-4-22

Strengths

  1. The paper provides a comprehensive overview of the impact of different sampling algorithms for diffusion based generative models. The authors assess how these choices impact sample quality and generation time.

  2. The authors show significant speedups in generation time with good quality as compared to the baseline sampling methods used in prior works

Report

The authors have improved the clarity and presentation of results from the previous version. They have also provided a more comprehensive study on the JetNet dataset adds strength to the paper. They have addressed the feedback from the first draft and I I am happy to recommend publication.

Recommendation

Publish (meets expectations and criteria for this Journal)

---

## Round 2 · Referee Report · Anonymous (Referee 3) · 2025-5-2

Report

I am satisfied with the authors' changes and recommend the manuscript proceed to publication.

Recommendation

Publish (meets expectations and criteria for this Journal)

---

## Round 2 · List of Changes

A set of changes is made:

  1. Adding a training detail subsection, as well as more training details in each experiments (e.g. uncertainties, detailed setups...)
  2. Adjusting a big bunch of captions, formulas and plots
  3. Rephrase sentences

---

## Editorial Decision

published